# DATA-DRIVEN HIGHER ORDER DIFFERENTIAL EQUATIONS INSPIRED GRAPH NEURAL NETWORKS

**Moshe Eliasof**
Department of Applied Mathematics and Theoretical Physics
University of Cambridge
me532@cam.ac.uk

**Eldad Haber**
Department of Earth, Ocean and Atmospheric Sciences
University of British Columbia
ehaber@eoas.ubc.ca

**Eran Treister**
Computer Science Department
Ben-Gurion University of the Negev
erant@cs.bgu.ac.il

**Carola-Bibiane Schönlieb**
Department of Applied Mathematics and Theoretical Physics
University of Cambridge
cbs31@cam.ac.uk

## ABSTRACT

A recent innovation in Graph Neural Networks (GNNs) is the family of Differential Equation-Inspired Graph Neural Networks (DE-GNNs), which leverage principles from continuous dynamical systems to model information flow on graphs with built-in properties. However, existing DE-GNNs rely on first or second-order temporal orders. In this paper, we propose a neural extension to those pre-defined temporal dependencies. We show that our model, called TDE-GNN, can capture a wide range of temporal dynamics that go beyond typical first or second-order methods, and provide use cases where existing temporal models are challenged. We demonstrate the benefit of learning the temporal dependencies using our method rather than using pre-defined temporal dynamics on several graph benchmarks.

## 1 INTRODUCTION

In recent years, it has been shown that GNNs can be viewed as the discretization of Ordinary Differential Equations (ODE), providing interpretable behavior, such as smoothing (Poli et al., 2019; Chamberlain et al., 2021), energy conservation (Eliasof et al., 2021; Rusch et al., 2022), anti-symmetry (Gravina et al., 2023; 2024), pattern formation (Wang et al., 2022; Choi et al., 2023b), and more. We refer to this family of architectures as DE-GNNs. Many works in the field of DE-GNNs consider *stationary data*, that is, data that is not time dependent. As such, most works focus on the *spatial* interactions between nodes, while employing first-order temporal dynamics. As we show in Example 1, this can be limiting. Therefore, we study the importance of the higher-order DE-GNNs, and propose a novel mechanism to model the of the underlying ODEs of GNN layers in a data-driven fashion.

*The goal of this work is to develop and study a novel mechanism to model the temporal domain of DE-GNNs.* Our approach, called *TDE-GNN* , is based on learning (i) the *temporal order*, and, (ii) the *temporal dependency* in a data-driven fashion. The temporal order defines the order of the underlying ODE as favored by the data, while the temporal dependency specifies the relationship between intermediate DE steps. To the best of our knowledge, this is the first work to study the temporal domain of DE-GNNs, in the sense that it learns higher-order DEs in a general manner. All DE-GNNs known to us, utilize either first or second-order time dependencies. In other words, existing DE-GNNs assume fixed temporal behavior, which is constant in time and is pre-defined. This shortcoming, as we show later, can be rather limiting when complex phenomena are to be modeled. Our TDE-GNN can also be viewed as an extension for Residual Networks (He et al., 2016), which is aimed to incorporate node features from previous time steps (i.e., layers) in a learnable manner.

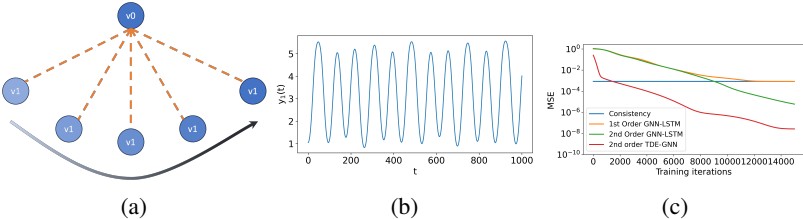

| (a) | (b) | (c) |

Figure 1: The pendulum location prediction example. (a) Illustration of a Pendulum's motion (b). Pendulum $y_1(t)$ coordinate vs. time (c) The prediction performance of naive, 1st, and 2nd order models. Higher-order models offer improved predictions.

## 2 MATHEMATICAL BACKGROUND AND MOTIVATION

**Notations.** We consider a graph $G = (V, E)$, where $V$ is a set of $n$ nodes, and $E \subseteq V \times V$ is a set of $m$ edges. The $i$-th node is associated with a possibly time-dependent hidden feature vector $f_i(t) \in \mathbb{R}^k$. Let $F(t) = [f_0(t), \dots, f_{n-1}(t)]^\top$ be a matrix of the node state (features) at time $t$.

**Differential Equations Inspired GNNs (DE-GNNs).** The basic idea of the DE-GNN family of architectures is to view GNNs as the discretization of the following ODE:

$$\frac{\partial F}{\partial t} = s\left(F(t); G\right), \tag{1}$$

with initial condition $F(t = 0) = \mathbf{F}^{(0)}$, and $s(F(t); G)$ is a spatial operator that depends on the graph $G$ and the node features $F(t)$. Specifically, it is common to employ graph diffusion, combined with a channel mixing operator implemented by a multilayer perceptron (MLP). Some examples of such methods were proposed in Chamberlain et al. (2021); Eliasof et al. (2021), and others. Because we focus on the *temporal* component of the ODE in this paper, we employ a similar spatial term that combines diffusion and channel mixing, as discussed later. Then, the graph ODE in Equation (1) is discretized in time, until time $T$, typically with the forward Euler method. The chosen discretization times are considered as GNN layers, with a total of $L$ time steps with step size $h$ such that $T = hL$.

**Order Matters.** The tasks considered in this work predict node values. The common theme is that we view them as the prediction of the time and space evolution of the node features, given past and current node features. A popular approach is to treat the problem by combining a GNN with time series mechanisms such as LSTM (Hochreiter & Schmidhuber, 1997) or GRU (Cho et al., 2014). While such techniques have shown promising results, we provide Example 1, where a standard GNN-LSTM is challenged, in the sense that it does not perform better than a naive solution. We attribute this shortcoming to the basic assumption of models like LSTM, that future predictions can be based on the previous state, effectively assuming first-order dynamics, which may not be sufficient to model higher-order phenomena, as shown below.

**Example 1.** (Nonlinear Pendulum) *We consider the problem of a nonlinear pendulum as a graph with two nodes, $v_0$ fixed at $(0, 0)$, and, $v_1$, located at $(x_1(t), y_1(t))$, illustrated in Figure 1a. Evaluating the coordinates of node $v_1$ at time $t$ can be done by solving the Newtonian mechanics that define the pendulum's motion:*

$$\frac{\partial^2 F}{\partial t^2} = q(F), \tag{2}$$

*where $q(F)$ is a gradient of the energy that characterizes the behavior of the pendulum. We discretize Equation (2) using the leapfrog method (Ascher, 2008), to generate a time series data of the pendulum vertices locations. Recall that node $v_0$ is static, and remains in (0,0), while $v_1$ moves according to Equation (2). We plot the location of $v_1$ in Figure 1b. We define a task whose inputs are observed locations of the pendulum nodes, and the goal is to predict the future locations of nodes. We consider four possible prediction models to address this task: (i) A* naive *prediction model, oftentimes called a* consistency *model, that simply outputs the latest available state. Formally, it is given by $\mathbf{F}^{(l+1)} = \mathbf{F}^{(l)}$. (ii) A GNN-LSTM, similar to Seo et al. (2018), (iii) a second order GNN-LSTM, and, (iv) our TDE-GNN limited to second order for a fair comparison, to be defined later in Section 3.*

*We report the obtained prediction mean squared error (MSE) compared to the ground-truth data in Figure 1c. We observe that the first-order GNN-LSTM model is as limited as the naive model of consistency. To understand the limitations of GNN-LSTM, it is key to recall Equation (2), and see that a pendulum's motion involves a second-order system. However, the first-order GNN-LSTM mechanism considers only the latest state (node features) $\mathbf{F}^{(l)}$. Indeed, when considering a second-order GNN-LSTM, one can obtain improved performance. Finally, we see that a second-order TDE-GNN offers further prediction performance improvement, as plotted in Figure 1c.*

*This example demonstrates that the order (length) of the history used for prediction is important. If too short of a history is used, it may be impossible to accurately predict the behavior of systems with order higher than the available history length. Since for many problems, the order is unknown and can vary in time, we allow TDE-GNN to learn the order from the data.*

## 3 LEARNING HIGHER-ORDER DE-GNNS

As discussed, previous works have so far mostly considered first order time dependent GNNs as in Equation (1), whose forward Euler discretization reads:

$$\mathbf{F}^{(l+1)} = \mathbf{F}^{(l)} + hs(\mathbf{F}^{(l)}; G), \tag{3}$$

where $\mathbf{F}^{(l)} \in \mathbb{R}^{n \times k}$ are the node features at the $l$-th layer, $h$ is a positive step size, and $s$ is the spatial term specified $s$ in Appendix F.2. Here, we study the temporal dynamics, and introduce a TDE-GNN layer that stems from the following ODE, with a maximal order of the hyperparameter $o \geq 1$:

$$\sum_{p=1}^{o} c_p \frac{\partial^p F}{\partial t^p} = s\left(F(t); G\right), \tag{4}$$

accompanied by the initial conditions

$$\left.\frac{\partial^p F}{\partial t^p}\right|_{t=0} = F^{(p)}(t=0) \quad p = 0, \dots, o-1. \tag{5}$$

The forward Euler discretization of Equation (4) yields our TDE-GNN layer:

$$\mathbf{F}^{(l+1)} = \sum_{p=1}^{o} c_p(\mathcal{H}_o^{(l)})\mathbf{F}^{(l-p+1)} + hs(\mathbf{F}^{(l)}; G). \tag{6}$$

Here, we define $\mathbf{c}(\mathcal{H}_o^{(l)}) = [c_1(\mathcal{H}_o^{(l)}), \dots, c_o(\mathcal{H}_o^{(l)})] \in \mathbb{R}^o$, which are learned weights based on previous node features of up to order $o$, formally denoted by:

$$\mathcal{H}_o^{(l)} = [\mathbf{F}^{(l)} \| \mathbf{F}^{(l-1)} \| \dots \| \mathbf{F}^{(l-o+1)}] \in \mathbb{R}^{o \times n \times k}, \tag{7}$$

where $\|$ denotes the stacking operation. It is important to note that Equation (6) models the dynamics of the $l$-th layer with $o-1$ previous layers, and therefore yields a discretization of an ODE of order $o$. For stability of computation, and for interoperability, we demand that $\sum_{p=1}^{o} c_p = 1$. This constraint ensures that the coefficients approximate derivatives of the data up to $o$-th order (Evans, 1998). In Appendix C we describe the implementation of learning the temporal coefficients $\mathbf{c}(\mathcal{H}_o^{(l)})$.

**Understanding the learned coefficients $\mathbf{c}(\mathcal{H}_o^{(l)})$.** It is important to note that Equation (6) extends the idea of residual networks. Typical residual networks (as in ResNet (He et al., 2016)) can be obtained by Equation (6) with $o = 1$ and $c_1 = 1$, which yields the forward Euler method that is often used to discretize diffusive GNNs (Chamberlain et al., 2021; Eliasof et al., 2021). Also, second order oscillatory GNNs as proposed in Eliasof et al. (2021); Rusch et al. (2022), can be implemented by Equation (6) with $o = 2$ and $\mathbf{c} = [c_1, c_2] = [2, -1]$. Overall, our TDE-GNN can implement, as well as extend, both of these types of architectures by learning higher-order dynamics with adaptive coefficients $\mathbf{c}(\mathcal{H}_o^{(l)})$. Those extensions allow our TDE-GNN to model a diverse family of dynamics that cannot be obtained with the aforementioned methods. We provide Example 2 that shows how a third-order DE-GNN is implemented by our method. Thus, treating the temporal domain in DE-GNNs using our learnable framework allows us to *reveal and understand the order of the underlying time-dependent process in a data-driven fashion*. Furthermore, as shown in Example 1, such a treatment can be crucial to accurately model data that stems from higher-order phenomena.

## 4 EXPERIMENTS

To demonstrate the efficacy of TDE-GNN, we experiment with two tasks: (i) node classification reported in Appendix D, and, (ii) spatio-temporal node forecasting, on several benchmarks. We provide benchmark details and statistics in Appendix G.2. The hyperparameters are determined using a grid search, as discussed in Appendix G.3. Because we propose two possible implementations of the temporal learning mechanism in Section Appendix C, we denote the *direct* parameterization variant by TDE-GNN$_D$ and the *attention* based parameterization variant by TDE-GNN$_A$. A detailed description of the TDE-GNN architectures is given in Appendix F.3. We also experiment with a *baseline* model that we call DE-GNN and is implemented according to Equation (6) with $o = 1$ and $c_1 = 1$, that is, it considers only first-order dynamics, similarly to existing GNNs inspired by DEs. Our code is available at `https://github.com/MosheEliasof/TDE-GNN`.

**Spatio-Temporal Forecasting** We experiment with spatio-temporal datasets, where the goal is to forecast future node values, given time-series data. We use the Chickenpox-Hungary, PedalMe-London, and Wikipedia-Math datasets from Rozemberczki et al. (2021b). We use incremental training, mean-squared-error (MSE) loss, and testing procedure from Rozemberczki et al. (2021b). We report the prediction performance, in terms of MSE, in Table 1, considering recent methods like DCRNN (Li et al., 2018), GConv (Seo et al., 2018), GC-LSTM (Chen et al., 2018a), DyGrAE (Taheri et al., 2019; Taheri & Berger-Wolf, 2019), EGCN Pareja et al. (2020), A3T-GCN (Zhu et al., 2020a), T-GCN (Zhao et al., 2019), MPNN LSTM (Panagopoulos et al., 2021), and AGCRN (Bai et al., 2020). Our results are reported in Table 1, showing improvement over existing temporal GNN models, as well as the baseline of DE-GNN. This result further highlights the importance of learning higher-order dynamics offered by our TDE-GNN.

| Dataset | Chickenpox Hungary | PedalMe London | Wikipedia Math |
|---|---|---|---|
| **Temporal GNNs** | | | |
| DCRNN | 1.124±0.015 | 1.463±0.019 | 0.679±0.020 |
| GConvGRU | 1.128±0.011 | 1.622±0.032 | 0.657±0.015 |
| GC-LSTM | 1.115±0.014 | 1.455±0.023 | 0.779±0.023 |
| DyGrAE | 1.120±0.021 | 1.455±0.031 | 0.773±0.009 |
| EGCN-O | 1.124±0.009 | 1.491±0.024 | 0.750±0.014 |
| A3T-GCN | 1.114±0.008 | 1.469±0.027 | 0.781±0.011 |
| T-GCN | 1.117±0.011 | 1.479±0.012 | 0.764±0.011 |
| MPNN LSTM | 1.116±0.023 | 1.485±0.028 | 0.795±0.010 |
| AGCRN | 1.120±0.010 | 1.469±0.030 | 0.788±0.011 |
| **Vanilla baseline** | | | |
| DE-GNN | 0.998±0.022 | 1.329±0.041 | 0.714±0.019 |
| **TDE-GNN (ours)** | | | |
| TDE-GNN$_D$ | 0.792±0.028 | 1.096±0.057 | 0.614±0.023 |
| TDE-GNN$_A$ | 0.787±0.018 | 0.714±0.051 | 0.565±0.017 |

Table 1: The performance of spatio-temporal networks evaluated by the average MSE and standard deviation ($\downarrow$) of 10 experimental repetitions.

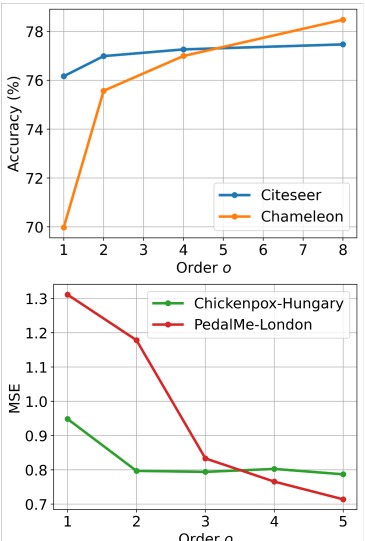

Figure 2: The impact of the model order $o$ on the performance of TDE-GNN.

**The influence of the order $o$.** As shown in Example 1, having sufficiently high order can be crucial to modeling complex data. In that example, we have demonstrated this significance via a synthetic task where the order is known. We now supplement this study by reporting the obtained performance as a function of the order $o$ on real-world datasets, where the exact order of the underlying process that generated the data is unknown. To provide a comprehensive study of the impact of $o$, we report the results on both the node classification and spatio-temporal node forecasting tasks considered in this work, on several datasets. The results are reported in Figure 2. Our results indicate, in congruence with Example 1, that higher-order models can improve performance compared to using a first-order model only. We also note that interestingly, for the Chickenpox-Hungary dataset, we see that a second-order model performs almost as well as a third, fourth, or fifth-order model. While we do not know the exact order of such a real-world dataset, the empirical results may hint at the actual underlying process of the spread of the Chickenpox disease in this dataset being second order.

**Inspecting $\mathbf{c}(\mathcal{H}_o^{(l)})$.** In Example 1, the underlying process of the data is known to be second-order, we now inspect and analyze the obtained coefficients $\mathbf{c}$ for a varying order $o \in \{2, 3, 4, 5\}$. As we show in Appendix G.5, it is possible to verify that the learned coefficients yield a valid discretization of the second-derivative operator $\frac{\partial^2 F}{\partial t^2}$, revealing the true order of the pendulum's motion.

## 5 CONCLUSIONS

In this paper, we showed that incorporating higher-order interactions can be crucial to model data that arises from complex phenomena. This understanding motivated us to develop a novel architecture called TDE-GNN, that offers the modeling and understanding of higher-order ODE behaviors in GNNs, as well as improved downstream task performance.

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

## A    RELATED WORK

**Graph Neural Networks Inspired by Differential Equations.** Adopting the interpretation of convolutional neural networks (CNNs) as discretizations of ODEs and PDEs (Ruthotto & Haber, 2018; Chen et al., 2018b; Zhang et al., 2019) to GNNs, works like GCDE (Poli et al., 2019), GODE (Zhuang et al., 2020), GRAND (Chamberlain et al., 2021), PDE-GCN$_D$ (Eliasof et al., 2021), GRAND++ (Thorpe et al., 2022) and others, propose to view GNN layers as time steps in the integration of the non-linear heat equation. This perspective allows to control the diffusion (smoothing) in the network, to understand oversmoothing (Nt & Maehara, 2019; Oono & Suzuki, 2020; Cai & Wang, 2020) in GNNs. Thus, works like Chien et al. (2021); Luan et al. (2022); Giovanni et al. (2023) propose to utilize a *learnable* diffusion term, thereby alleviating oversmoothing. Other architectures like PDE-GCN$_M$ (Eliasof et al., 2021) and GraphCON (Rusch et al., 2022) propose to mix diffusion and oscillatory processes (e.g., based on the wave equation) to avoid oversmoothing by introducing a feature energy preservation mechanism. Nonetheless, as noted in Rusch et al. (2023), besides alleviating oversmoothing, it is also important to design GNN architectures with improved expressiveness. Recent examples of such networks are Gravina et al. (2023; 2024) that propose an anti-symmetric GNN to alleviate over-squashing (Alon & Yahav, 2021), Wang et al. (2022); Choi et al. (2023b) that formulate a reaction-diffusion GNN to enable non-trivial pattern growth, Zhao et al. (2023) that propose a convection-diffusion based GNN, advection-reaction-diffusion to allow directed information transportation (Eliasof et al., 2023), and Maskey et al. (2023) that formalize a fractional Laplacian ODE based GNN with improved expressiveness. A common theme of most of the aforementioned works, is the focus on the *spatial* term of the ODE, while the temporal term is set to be of first or second order. In this work, we propose to extend the family of ODE-inspired GNNs from the perspective of the *temporal* domain.

**The Temporal Domain in Graph Neural Networks.** In recent years, GNNs for spatio-temporal data were developed. Some examples are Chen et al. (2018a); Seo et al. (2018); Zhao et al. (2019) that combine graph convolution with LSTM mechanisms, and other combines graph attention with temporal mechanisms, as in Zhu et al. (2020a). Other works like Pareja et al. (2020); Bai et al. (2020) propose adaptive graph convolutions for temporal graphs. It has also been shown in Gutteridge et al. (2023) that adjacency matrix update according to intermediate node features is useful for long-range benchmarks. Furthermore, recent works have shown that GNNs for temporal graph datasets can benefit from the interpretation and construction of ordinary differential equations. For example, it was shown in Xiong et al. (2023); Sun et al. that reaction and diffusion systems can improve traffic prediction, and it was shown in Choi et al. (2023a) that advection and diffusion can improve weather forecasting performance. However, all the considered works discussed here utilize first-order temporal dynamics, while focusing on the spatial term of the ordinary differential equation. In this paper, we explore and study the temporal domain in the context of DE-GNNs, and show its importance to model complex systems and improve performance.

## B    3RD ORDER MODEL BY TDE-GNN

**Example 2.** (3rd Order TDE-GNN) *We now draw a link between a third-order TDE-GNN (with an order hyperparameter $o = 3$), and a third-order ODE. Note that every set of coefficients $\{c_1, c_2, c_3\}$ that sum to 1, with a step size $h = 1$ can be spanned by the basis:*

$$\begin{pmatrix} c_1 \\ c_2 \\ c_3 \end{pmatrix} = \alpha_1 \begin{pmatrix} 1 \\ 0 \\ 0 \end{pmatrix} + \alpha_2 \begin{pmatrix} 2 \\ -1 \\ 0 \end{pmatrix} + \alpha_3 \begin{pmatrix} 2 \\ -2 \\ 1 \end{pmatrix}, \tag{8}$$

*with the constraint $\sum_{i=1}^{3} \alpha_i = 1$. Note that the vectors that multiply $\alpha_1, \alpha_2$ and $\alpha_3$ correspond to a first-order, second-order, and third-order finite difference, respectively (i.e., $\frac{\partial F(t)}{\partial t}$, $\frac{\partial^2 F(t)}{\partial t^2}$, and $\frac{\partial^3 F(t)}{\partial t^3}$). Since the basis in Equation (8) is complete, for any $c_1, c_2, c_3$ that sum to 1, there exists a 3rd order differential equation whose discretization yields the same coefficients.*

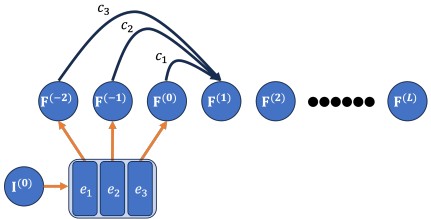

Figure 3: The embedding of input features $\mathbf{I}$ using an MLPs $(e_1, e_2, e_3)$ to obtain $o = 3$ initial conditions, followed by our TDE-GNN for stationary problems.

## C   IMPLEMENTING $\mathbf{c}(\mathcal{H}_o^{(l)})$

At the core of our TDE-GNN stands the learning of the temporal coefficients $\mathbf{c}(\mathcal{H}_o^{(l)})$, with two key requirements for a valid implementation: (i) the vector $\mathbf{c}$ sums to 1, i.e., $\sum_{p=1}^{o} c_p(\mathcal{H}_o^{(l)}) = 1$, and, (ii) the entries of $\mathbf{c}(\mathcal{H}_o^{(l)})$ can be any real-valued number. These requirements offer both training stability (due to the normalization in requirement (i)), and the approximation of a finite order derivative (Evans, 1998). We now discuss two implementations that we consider in this paper, and later, in our experiments in Section 4, we compare their performance.

**Direct parameterization.** Perhaps the most intuitive implementation is obtained by *direct* parameterization, where we directly learn a vector $\tilde{\mathbf{c}} \in \mathbb{R}^o$, and divide it by the sum of its entries (to satisfy requirement (i)), leading to the temporal coefficients vector:

$$\mathbf{c} = \frac{\tilde{\mathbf{c}}}{\sum_{p=1}^{o} \tilde{c}_p}. \tag{9}$$

Note that in this case, the coefficients vector $\mathbf{c}$ is not directly influenced by the history $\mathcal{H}_o^{(l)}$, but it is still optimized according to the history via backpropagation.

**Attention-based parameterization.** In addition to the direct parameterization, we propose a novel mechanism that leverages an attention mechanism as in Vaswani et al. (2017). A key feature of the attention mechanism is that it outputs a pairwise score of its input. The *novelty* here is to apply the attention mechanism on the *temporal* dimension. To this end, by collecting and appropriately shaping the node features of the previous $o$ layers, denoted by $\mathcal{H}_o^{(l)} \in \mathbb{R}^{o \times n \times k}$ as defined in Equation (6), and feeding it to an attention layer (Vaswani et al., 2017), one obtains a pairwise score map $\mathcal{S} \in [0, 1]^{o \times o}$. The last row in $\mathcal{S}$ represents the temporal scores of the $l$-th layer with the previous layers $o - 1$. Clearly, requirement (i) is met by the SoftMax function used in the attention mechanism in Vaswani et al. (2017). However, a SoftMax function yields non-negative pairwise values, which do not satisfy requirement (ii). Such a limitation will prevent, for instance, the ability to implement the oscillatory equation discussed in 3 using our TDE-GNN. Therefore, we follow the same implementation as in Vaswani et al. (2017) up to the SoftMax step. Instead, we only apply a normalization of the obtained pairwise interaction map by dividing it by its sum. This procedure satisfies both requirements (i) and (ii). We provide further details about the implementation in Appendix F.1.

**Initial conditions.** When considering high-order ODEs, the aspect of the initial conditions of the model is important (Ascher & Petzold, 1998). We consider two use cases that are treated differently. First, when solving a stationary problem such as node classification, where only a single initial temporal condition is available, we use $o$ MLPs to embed this single state into $o$ states, and then use the network in Equation (6). This initialization, as well as the application of a TDE-GNN at the first layer, is illustrated in Figure 3.

For stationary problems (e.g., node classification), the input consists of a single time step at time $T_0$. Given input features $\mathbf{I}^{(0)} \in \mathbb{R}^{n \times k_{in}}$, we embed them using an MLP to obtain the initial conditions of the ODE, denoted by $\mathbf{F}^{(0)} \in \mathbb{R}^{n \times k}$. For spatio-temporal tasks (e.g., forecasting node quantities), the node features $[\mathbf{I}^{(0)}, \ldots, \mathbf{I}^{(r)}]$ are provided at sampled times $[T_0, \ldots, T_r]$, and embedded in latent space. The node features at the final GNN layer $\mathbf{F}^{(L)}$ are then fed to a classifier to output the desired shaped prediction to be compared with the labeled data, depending on the task. Note that for time

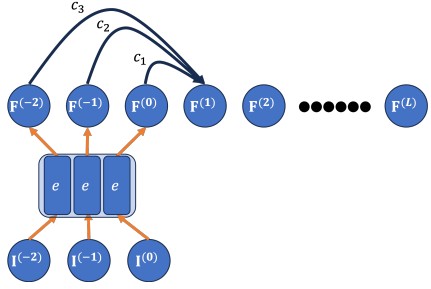

Figure 4: The initialization of TDE-GNN for spatio-temporal data with a history of $o = 3$.

series graph problems where we have a time series as input, we use at least $o$ historical data in order to initialize the states. In this case, the frequency of the *observed input* can be different than the frequency of the hidden space $\mathbf{F}^{(l)}$ that discretizes the ODE, in the sense that we can use more hidden layers than observed inputs. Upon receiving at least $o$ input observations, we embed them using an MLP to obtain $o$ hidden initial conditions. In Figure 4, we illustrate the described process as well as the application of a TDE-GNN layer to the inputs. Past the initialization step, in both stationary and non-stationary cases, the features update relies on previously computed hidden node features, as described in Equation (6).

**Complexity.** Compared to existing DE-GNN methods, our TDE-GNN involves additional $o - 1$ additions of previous node features, to allow modeling differential equations of order $o$, and achieve improved performance, as we show in Section 4. If the coefficients $\mathbf{c}$ are obtained using the *direct* parameterization, then $o - 1$ scalar multiplications are required. If the *attention* based mechanism is utilized to learn and evaluate $\mathbf{c}$, adding $\mathcal{O}(n \cdot k \cdot o^2)$ multiplications. We note that $o$, the order hyperparameter of TDE-GNN, is typically significantly smaller than the number of channels $k$ and nodes $n$, because it is bounded by the number of layers $L$. In Appendix G.4 we report the training and inference runtimes of the proposed implementations.

**Properties of TDE-GNN.** Our TDE-GNN draws inspiration from a stable discrete process of ODE integration, that generates future time values, which is regarded as the node features evolution throughout the layers. Therefore, a natural question that arises is whether the obtained network is stable. Indeed, if the proposed architecture is unstable, then it may be difficult to fit the data, or, the network may not generalize well (see Haber & Ruthotto (2017) for stability definition and a thorough discussion). To this end, we prove the following theorem in Appendix E.

**Theorem 1** *(Stability of TDE-GNN). For the discretization of Equation* (6)*, there exists a vector* $\mathbf{c} = [c_1, \ldots, c_o]$ *such that the discrete solution is stable.*

In our study in Section 4, we verify Theorem 1, and show that the learnable weights $\mathbf{c}$ can be interpreted as finite difference derivatives.

## D   NODE CLASSIFICATION EXPERIMENTAL RESULTS

We experiment with homophilic and non-homophilic datasets. The homophilic datasets are Cora (McCallum et al., 2000), Citeseer (Sen et al., 2008), and Pubmed (Namata et al., 2012). The non-homophilic datasets are Chameleon, Squirrel, and Film from Rozemberczki et al. (2021a). In all cases, we use the 10 splits from Pei et al. (2020), and report their average accuracy and standard deviation in Table 2. We consider three types of baselines: (i) 'general' GNN architectures, such as GCN (Kipf & Welling, 2017), GAT (Veličković et al., 2018), GCNII (Chen et al., 2020), Geom-GCN (Pei et al., 2020), NSD (Bodnar et al., 2022), GGCN (Yan et al., 2021), H2GCN (Zhu et al., 2020b), FAGCN (Bo et al., 2021), GPRGNN (Chien et al., 2021), GRAFF (Giovanni et al., 2023), LINKX (Lim et al., 2021), and ACMII (Luan et al., 2022). (ii) GNNs inspired by differential equations (DEs), including: GRAND (Chamberlain et al., 2021), PDE-GCN (Eliasof et al., 2021), GRAND++ (Thorpe et al., 2022), GREAD (Choi et al., 2023b), CDE (Zhao et al., 2023), and FLODE (Maskey et al., 2023). The common theme of those methods is that all consider first-order temporal behavior with $o = 1, c_1 = 1,$

except for PDE-GCN, which considers a second-order model, where $o = 2$, $\mathbf{c} = [c_1, c_2] = [2, -1]$. In contrast, our TDE-GNN can learn the vector of coefficients $\mathbf{c}$ with maximal order $o$, and unless otherwise specified, we set the order to be the number of layers in the network, i.e., $o = L$. The third baseline we consider is (iii) a vanilla version of our TDE-GNN, where $o = 1, c_1 = 1$, and we call this variant DE-GNN. Our results in Table 2 suggest that for homophilic graphs which are known to benefit from diffusion (Gasteiger et al., 2019), TDE-GNN performs similarly to other first-order differential equations inspired GNNs, including our baseline DE-GNN, since a diffusion process can be described using a first-order ODE. We find that the significance of learning higher-order dynamics is more pronounced for non-homophilic graphs which may stem from more complex phenomena than homophilic graphs. For instance, we find that our TDE-GNN$_A$ achieves an accuracy of 78.48%, compared to the baseline vanilla, first order DE-GNN with 70.99% – a considerable improvement.

| Dataset | Squirrel | Film | Chameleon | Citeseer | Pubmed | Cora |
|---|---|---|---|---|---|---|
| Homophily | 0.22 | 0.22 | 0.23 | 0.71 | 0.74 | 0.81 |
| **General GNNs** | | | | | | |
| GCN | 23.96±2.01 | 26.86±1.10 | 28.18±2.24 | 73.68±1.36 | 88.13±0.50 | 85.77±1.27 |
| GAT | 30.03±1.55 | 28.45±0.89 | 42.93±2.50 | 74.32±1.23 | 87.62±1.10 | 86.37±0.48 |
| GCNII* | 38.47±1.58 | 32.87±1.30 | 60.61±3.04 | 77.13±1.48 | 90.30±0.43 | 88.49±1.25 |
| Geom-GCN* | 38.32±0.92 | 31.63±1.15 | 60.90±2.81 | 77.99±1.15 | 90.05±0.47 | 85.27±1.57 |
| NSD* | 56.34±1.32 | 37.79±1.15 | 68.68±1.58 | 77.14±1.57 | 89.49±0.40 | 87.14±1.13 |
| GGCN | 55.17±1.58 | 37.81±1.56 | 71.14±1.84 | 77.14±1.45 | 89.15±0.37 | 87.95±1.05 |
| H2GCN | 36.48±1.86 | 35.70±1.00 | 60.11±1.71 | 77.11±1.57 | 89.49±0.38 | 87.87±1.20 |
| FAGCN | 42.59±0.69 | 34.87±1.35 | 55.22±2.11 | 74.01±1.85 | 76.57±1.88 | 86.34±0.67 |
| GPRGNN | 31.61±1.24 | 34.63±1.22 | 46.58±1.71 | 77.13±1.67 | 87.54±0.38 | 87.95±1.18 |
| GRAFF* | 59.01±1.31 | 37.11±1.08 | 71.38±1.47 | 77.30±1.85 | 90.04±0.41 | 88.01±1.03 |
| LINKX | 61.81±1.80 | 36.10±1.55 | 68.42±1.38 | 73.19±0.99 | 87.86±0.77 | 84.64±1.13 |
| ACMII* | 67.40±2.21 | 37.09±1.32 | 74.76±2.20 | 77.12±1.58 | 89.71±0.48 | 88.25±0.96 |
| **GNNs Inspired by DEs** | | | | | | |
| GRAND | 40.05±1.50 | 35.62±1.01 | 54.67±2.54 | 76.46±1.77 | 89.02±0.51 | 87.36±0.96 |
| PDE-GCN* | N/A | N/A | 66.01±2.11 | 78.45±1.98 | 89.93±0.62 | 88.60±1.77 |
| GRAND++ | 40.06±1.70 | 33.63±0.48 | 56.20±2.15 | 76.57±1.46 | 88.50±0.35 | 88.15±1.22 |
| GREAD* | 59.22±1.44 | 37.90±1.17 | 71.38±1.30 | 77.60±1.81 | 90.23±0.55 | 88.57±0.66 |
| CDE* | 55.04±1.73 | 40.08±1.49 | 68.45±2.47 | 80.04±1.75 | 90.05±0.64 | 87.19±1.44 |
| FLODE | 64.23±1.84 | 37.16±1.42 | 73.60±1.55 | 78.07±1.62 | 89.02±0.38 | 86.44±1.17 |
| **Vanilla baseline** | | | | | | |
| DE-GNN | 63.97±1.77 | 36.04±1.08 | 70.99±2.27 | 76.58±1.89 | 89.92±0.59 | 87.03±1.14 |
| **TDE-GNN (ours)** | | | | | | |
| TDE-GNN$_D$ | 70.19±1.74 | 37.29±1.19 | 77.38±2.05 | 77.66±1.91 | 90.28±0.53 | 87.99±1.02 |
| TDE-GNN$_A$ | 71.38±1.93 | 37.02±1.27 | 78.48±2.11 | 77.47±1.82 | 90.08±0.49 | 87.93±0.95 |

Table 2: Node classification accuracy (%). $\uparrow$. * denotes the best result out of several variants.

# E  PROOF TO THEOREM 1

We now prove Theorem 1 from the main paper.

**Proof:** Let us write the discrete DE in Equation (6) explicitly as

$$\mathbf{F}^{(l+1)} = c_o \mathbf{F}^{(l-o+1)} + \ldots c_1 \mathbf{F}^{(l)} + h s(\mathbf{F}^{(l)}; G). \tag{10}$$

Similar to the proofs for multistep ODE methods (Ascher et al., 1995), to prove the stability of the method, assuming the Jacobians of $s$ have non-positive real part[1], it is sufficient to consider only the temporal term:

$$\mathbf{F}^{(l+1)} = c_o \mathbf{F}^{(l-o+1)} + \ldots + c_1 \mathbf{F}^{(l)}$$

---

[1]If the Jacobian has a positive real part, then the underlying ODE is unstable, and therefore its discretization is also unstable.

This is a linear, constant-coefficient differential equation, and it must be stable for Equation (6) to be stable. Also, as common in proofs of multistep ODE methods (Ascher et al., 1995), we start from a solution of the form

$$\mathbf{F}^{(l)} = \xi^l$$

(meaning $\xi$ to the power of $l$). Substituting we obtain

$$\xi^{l+1} = c_o \xi^{l-o+1} + \ldots c_1 \xi^l$$

Dividing by $\xi^{l-o+1}$ we obtain the polynomial equation

$$\xi^{o+1} - c_1 \xi^o - \ldots - c_o = 0$$

This is a polynomial of degree $o + 1$ with coefficients $[1, -c_1, \ldots, -c_o]$ where $\sum c_i = 1$. Let $\rho(c)$ be the roots of the polynomial. It is straightforward to see (by substitution) that 1 is a root of the polynomial. For the solution to be stable, we need to have that:

$$\mathbf{F}^{(n)} = \xi^n$$
$$|\mathbf{F}^{(n)}| \le |\mathbf{F}^{(n-1)}|$$
$$|\xi^n| \le |\xi^{n-1}| \quad \to \quad |\xi| \le 1$$

In multi-step methods for ODEs, this condition is referred to as the root condition. Furthermore, as shown in Ascher et al. (1995), since the coefficients $c$ are to be determined (learned, in the case of TDE-GNN), there always exists a set of coefficients such that the root condition is satisfied. $\qquad\square$

**Remark 1** *While it is difficult to verify the root condition analytically, it is possible to compute it numerically, thus revealing the order of the process we learn, as we also show in G.5.*

## F  IMPLEMENTATION DETAILS

### F.1  IMPLEMENTING THE TEMPORAL TERM WITH ATTENTION

We now describe the implementation of TDE-GNN$_A$, i.e., the TDE-GNN implemented by an attention mechanism. Namely, to learn the dynamics between the node features at a current layer $l$ and the previous $o - 1$ layers, we utilize a multi-head self-attention mechanism (Vaswani et al., 2017) that assigns scores between the considered layer $l$ and the $o - 1$ layers. The difference in our implementation compared to a standard attention module as in Vaswani et al. (2017) is that we remove the SoftMax normalization step, as discussed in C, and it is required to allow both positive and negative numbers. We denote the attention mechanism by MHA. The MHA computes a score for each pair of layers $(l_i, l_j) \in o \times o$. As an input to the MHA, we use the history feature tensor as described in Equation (7), which is comprised of the stacking of the current and previous (layer-wise) $o - 1$ node features. Then, the output of the attention module is given by:

$$\tilde{\mathcal{S}}_o^{(l)} = \text{MHA}(\mathcal{H}_o^{(l)}) \in \mathbb{R}^{o \times o}. \tag{11}$$

The $(l_i, l_j)$-th entry in $\mathcal{S}_o^{(l)}$ represents the connection between the $l_i$-th and $l_j$-th layers. Specifically, the last row of $\mathcal{S}_o^{(l)}$ represents the connection between the current layer $l$ and the previous $o - 1$ layers. Therefore we define the unnormalized layer coefficients vector as the last row of $\tilde{\mathcal{S}}_o^{(l)}$, that can be extracted using the following Python notations:

$$\tilde{\mathbf{c}}(\mathcal{H}_o^{(l)}) = \tilde{\mathcal{S}}_o^{(l)}[-1, :] \in \mathbb{R}^o \tag{12}$$

In order to satisfy condition (i) that demands the weights to sum to 1, described in C, we also add a normalization step, such that the coefficients are defined as:

$$\mathbf{c}(\mathcal{H}_o^{(l)}) = \frac{\tilde{\mathbf{c}}(\mathcal{H}_o^{(l)})}{\sum \tilde{\mathbf{c}}(\mathcal{H}_o^{(l)})}. \tag{13}$$

### F.2 IMPLEMENTING THE SPATIAL TERM

The temporal mechanism developed in this paper is generic and can possibly be applied to various DE-GNNs, and is the novelty of our work. However, a GNN inspired by differential equations, and therefore also our TDE-GNN, is not complete without the spatial term that propagates node features across the graph. As discussed in Equation (1), our spatial aggregation function $s$ is based on the combination of a channel mixing operation realized by an MLP and the symmetric normalized graph Laplacian $\mathbf{L} = \mathbf{D}^{-\frac{1}{2}}(\mathbf{D} - \mathbf{A})\mathbf{D}^{-\frac{1}{2}}$ where $\mathbf{D}$ is the degree matrix, and $\mathbf{A}$ is the adjacency matrix of the graph G. Formally, the spatial aggregation is given by:

$$s(\mathbf{F}^{(l)}; G) = \sigma \left( \left( \mathbf{F}^{(l)} - h\mathbf{L}\mathbf{F}^{(l)} \right) \mathbf{W}^{(l)} \right), \tag{14}$$

where $\sigma$ is a non-linear activation function, ReLU in our implementation, and $h$ is a positive step size, and $\mathbf{W}^{(l)} \in \mathbb{R}^{k \times k}$ are the learnable weights of the MLP.

Therefore, substituting the prescribed temporal in Equation (14) into Equation (6) leads to our TDE-GNN layer, as follows:

$$\mathbf{F}^{(l+1)} = \sum_{p=0}^{o-1} c_p(\mathcal{H}_o^{(l)})\mathbf{F}^{(l-p)} + h\sigma \left( \left( \mathbf{F}^{(l)} - h\mathbf{L}\mathbf{F}^{(l)} \right) \mathbf{W}^{(l)} \right). \tag{15}$$

### F.3 ARCHITECTURE DETAILS

**Node classification architecture.** We now elaborate on the TDE-GNN architecture for stationary problems, as used in our node classification experiments. The overall architecture flow is similar to standard GNN architectures for node classification, such as GCN (Kipf & Welling, 2017) and GCNII (Chen et al., 2020). It is composed of initial embedding layers $e_1, \ldots, e_o$, followed by $L$ TDE-GNN layers, and a classifier implemented by a linear layer denoted by $e_{out}$. The complete flow of this architecture is described in Algorithm 1. We train this architecture on node classification datasets by minimizing the cross-entropy loss between the ground-truth node labels $\mathbf{Y}$ and the predicted node labels $\tilde{\mathbf{Y}}$.

---

**Algorithm 1** TDE-GNN for stationary problems with order $o$.

**Input:** Node features $\mathbf{I}^{(0)} \in \mathbb{R}^{n \times k_{in}}$
**Output:** Predicted node labels $\tilde{\mathbf{Y}} \in \mathbb{R}^{n \times k_{out}}$
1: **procedure** TDE-GNN
2: $\quad$ $\mathbf{I}^{(0)} \leftarrow \text{Dropout}(\mathbf{I}^{(0)}, p)$
3: $\quad$ $\mathbf{F}^{(-o+1)} = e_1(\mathbf{I}^{(0)}); \; \mathbf{F}^{(-o+2)} = e_2(\mathbf{I}^{(0)}); \ldots; \; \mathbf{F}^{(0)} = e_o(\mathbf{I}^{(0)})$
4: $\quad$ Initialize history tensor $\mathcal{H}_o^{(0)}$ according to Equation (7).
5: $\quad$ **for** $l = 0 \ldots L-1$ **do**
6: $\quad\quad$ $\mathbf{F}^{(l)} \leftarrow \text{Dropout}(\mathbf{F}^{(l)}, p)$
7: $\quad\quad$ Compute coefficients $\mathbf{c}(\mathcal{H}_0^{(l)})$ according to Appendix C.
8: $\quad\quad$ Update features $\mathbf{F}^{(l+1)}$ according to Equation (15).
9: $\quad\quad$ Update history tensor $\mathcal{H}_o^{(l+1)}$ according to Equation (7).
10: $\quad$ **end for**
11: $\quad$ $\mathbf{F}^{(L)} \leftarrow \text{Dropout}(\mathbf{F}^{(L)}, p)$
12: $\quad$ $\tilde{\mathbf{Y}} = e_{out}(\mathbf{F}^{(L)})$
13: $\quad$ Return $\tilde{\mathbf{Y}}$
14: **end procedure**

---

**Spatio-Temporal Node Forecasting.** The typical task in spatio-temporal datasets is to predict future quantities (e.g., driving speed) given several previous time steps (also called frames). Formally, one is given an input tensor $\mathbf{I}_{temporal}^{in} = [\mathbf{I}^{(0)}, \ldots \mathbf{I}^{(r)}] \in \mathbb{R}^{n \times rk_{in}}$, where $r$ is the number of input (observed) time frames, and the goal is to predict $a$ time frames ahead, i.e., the ground-truth is given by $\mathbf{I}_{temporal}^{gt} = [\mathbf{I}^{(r+1)}, \ldots, \mathbf{I}^{(r+a)}] \in \mathbb{R}^{n \times ak_{in}}$. This is in contrast to stationary datasets such as

Cora (McCallum et al., 2000), where input node features $\mathbf{I}^{(0)} \in \mathbb{R}^{n \times k_{in}}$ are given, and the goal is to fit to some ground-truth $\mathbf{Y} \in \mathbb{R}^{n \times k_{out}}$ which can also be of different dimensionality in its output space. In this context, a stationary dataset can be thought of as setting $r = a = 1$ for the non-stationary settings. We show the overall flow of our TDE-GNN architecture for non-stationary problems in Algorithm 2 [2].

In this architecture, we update the hidden state feature matrix $\mathbf{F}_{\text{state}}^{(l)}$ based on the hidden historical feature matrix $\mathbf{F}_{\text{hist}}^{(l)}$. The reason for this construction is that we want to continue from the current, most recent feature $\mathbf{F}_{\text{state}}^{(l)}$, but also consider the given historical data encoded in $\mathbf{F}_{\text{hist}}^{(l)}$.

Similarly to Attention models (Vaswani et al., 2017), we incorporate time embedding based on the concatenation of sine and cosine function evaluations with varying frequencies multiplied by the time of the input frames, as input to our TDE-GNN, denoted by $\mathbf{T}_{\text{emb}} \in \mathbb{R}^{n \times rk_{temb}}$, where we choose the number of frequencies to be 10, and by the concatenation of both sine and cosine lead to $k_{temb} = 20$. We note that the time embedding is computed in a pre-processing fashion. To initialize the hidden feature matrices $\mathbf{F}_{\text{state}}^{(0)}$, $\mathbf{F}_{\text{hist}}^{(0)}$, we embed the input data $\mathbf{I}_{\text{temporal}}^{in}$, concatenated with $\mathbf{T}_{\text{emb}}$, using two fully connected layers denoted by $e^{\text{state}}$ and $e^{\text{hist}}$.

During training, we minimize the mean squared error (MSE) between the ground truth future node quantities and the predicted quantities by TDE-GNN, similar to the training procedure of the rest of the considered methods in Table 1. Specifically, following Rozemberczki et al. (2021b), the goal is to predict the node quantities of the next time frame given 4 previous time frames.

---

**Algorithm 2** TDE-GNN for non-stationary problems with order $o$.

---

**Input:** Node features $\mathbf{I}_{temporal}^{in} = [\mathbf{I}^{(0)}, \dots \mathbf{I}^{(r)}] \in \mathbb{R}^{n \times rk_{in}}$, time embedding $\mathbf{T}_{\text{emb}} \in \mathbb{R}^{n \times rk_{temb}}$

**Output:** Predicted future node quantities $\tilde{\mathbf{I}}_{temporal}^{pred} = [\tilde{\mathbf{I}}^{(r+1)}, \dots, \tilde{\mathbf{I}}^{(r+a)}] \in \mathbb{R}^{n \times ak_{in}}$

1: **procedure** TDE-GNN
2:     $\mathbf{I}_{\text{temporal}}^{in} \leftarrow \text{Dropout}(\mathbf{I}_{\text{temporal}}^{in}, p)$
3:     $\mathbf{T}_{\text{emb}} \leftarrow \text{e}^{\text{time}-\text{embed}}(\mathbf{T}_{\text{emb}})$
4:     $\mathbf{F}_{\text{state}}^{(0)} = e^{\text{state}}(\mathbf{I}^{(r)} \oplus \mathbf{T}_{\text{emb}})$
5:     $\mathbf{F}_{\text{hist}}^{(0)} = e^{\text{hist}}(\mathbf{I}_{\text{temporal}}^{in} \oplus \mathbf{T}_{\text{emb}})$
6:     Initialize history tensor $\mathcal{H}_o^{(0)}$ according to Equation (7).
7:     **for** $l = 0 \dots L - 1$ **do**
8:         $\mathbf{F}_{\text{state}}^{(l)} \leftarrow \text{Dropout}(\mathbf{F}_{\text{state}}^{(l)}, p)$
9:         Compute coefficients $\mathbf{c}(\mathcal{H}_0^{(l)})$ according to Appendix C.
10:        Update features $\mathbf{F}_{state}^{(l+1)}$ according to Equation (15).
11:        Update history tensor $\mathcal{H}_o^{(l+1)}$ according to Equation (7).
12:        $\mathbf{F}_{\text{hist}}^{(l+1)} = e_l^{\text{hist}}(\mathbf{F}_{\text{hist}}^{(l)} \oplus \mathbf{F}_{\text{state}}^{(l+1)} \oplus \mathbf{T}_{\text{emb}})$
13:     **end for**
14:     $\mathbf{F}_{\text{state}}^{(L)} \leftarrow \text{Dropout}(\mathbf{F}_{\text{state}}^{(L)}, p)$
15:     $\tilde{\mathbf{Y}} = e_{out}^{\text{state}}(\mathbf{F}_{\text{state}}^{(L)})$
16:     Return $\tilde{\mathbf{I}}$
17: **end procedure**

---

# G EXPERIMENTAL DETAILS

## G.1 PENDULUM EXAMPLE PROBLEM

The pendulum's motion in Example 1 is modeled by a time-varying frequency pendulum, such that
$$q(F; t) = \sin(\omega(t)F),$$
where $\omega(t) = 1 - 0.04 \sin(t)$.

---

[2]In Algorithm 2, $\oplus$ denotes channel-wise concatenation.

## G.2    BENCHMARKS

**Node classification datasets.** We report the statistics of the datasets used in our node classification experiments in Table 3. All datasets are publicly available, and appropriate references to the data sources are provided in the main paper.

Table 3: Node classification datasets statistics.

| Dataset | Classes | Nodes | Edges | Features | Homophily |
|---------|---------|-------|-------|----------|-----------|
| Squirrel | 5 | 5,201 | 198,493 | 2,089 | 0.22 |
| Film | 5 | 7,600 | 33,544 | 932 | 0.22 |
| Chameleon | 5 | 2,277 | 36,101 | 2,325 | 0.23 |
| Citeseer | 6 | 3,327 | 4,732 | 3,703 | 0.80 |
| Pubmed | 3 | 19,717 | 44,338 | 500 | 0.74 |
| Cora | 7 | 2,708 | 5,429 | 1,433 | 0.81 |

**Spatio-temporal forecasting datasets.** We report the statistics of the datasets used in our spatio-temporal forecasting experiments in Table 4. All datasets are publicly available, and appropriate references to the data sources are provided in the main paper.

Table 4: Attributes of the spatio-temporal datasets, and information about the number of time periods ($T$) and spatial units ($|\mathcal{V}|$).

| Dataset | Frequency | $T$ | $|\mathcal{V}|$ |
|---------|-----------|-----|-----------------|
| Chickenpox Hungary | Weekly | 522 | 20 |
| Pedal Me Deliveries | Weekly | 36 | 15 |
| Wikipedia Math | Daily | 731 | 1,068 |

## G.3    HYPERPARAMETERS

All hyperparameters were determined by grid search, and the ranges and sampling mechanism distributions are provided in Table 5. Also, unless otherwise specified, in all experiments, we use $L = 8$ layers, and for node classification datasets, we use $o = L$. For the spatio-temporal datasets, we use $o = r$, i.e., the order is set to be equal to the number of historical data given by the task.

Table 5: Hyperparameter ranges

| Hyperparameter | Range | Uniform Distribution |
|----------------|-------|----------------------|
| input/output embedding learning rate | [1e-4, 1e-1] | log uniform |
| temporal term **c** learning rate | [1e-4, 1e-1] | log uniform |
| spatial term learning rate | [1e-4, 1e-1] | log uniform |
| input/output embedding weight decay | [0, 1e-2] | uniform |
| temporal term **c** weight decay | [0, 1e-2] | uniform |
| spatial term weight decay | [0, 1e-2] | uniform |
| input/output dropout | [0, 0.9] | uniform |
| hidden layer dropout | [0, 0.9] | uniform |
| use BatchNorm | { yes / no } | discrete uniform |
| step size h | [1e-3, 1] | uniform |
| hidden channels $k$ | { 8,16,32,64,128,256 } | discrete uniform |

## G.4    RUNTIMES

In addition to the complexity discussion in the main paper, we provide the measured runtimes in Table 6. Learning the temporal order and dynamics requires additional computations compared to

the vanilla baseline of DE-GNN, however, it also offers improved performance as we show in our experiments in Section 4. We report the measured training and inferences runtimes, and the number of parameters on the Cora dataset in Table 6. We measure the runtimes using an Nvidia-RTX3090 with 24GB of memory, which is the same GPU used to conduct our experiments.

Table 6: Training and inference GPU runtimes (milliseconds), and the number of parameters (thousands).

| Metric | DE-GNN ($o = 1, c_1 = 1$) | TDE-GNN$_D$ ($o = 8$) | TDE-GNN$_A$ ($o = 8$) |
|---|---|---|---|
| Training time | 21.45 | 23.96 | 34.55 |
| Inference time | 11.84 | 12.83 | 16.97 |
| Parameters | 125 | 157 | 174 |

### G.5 COEFFICIENTS ANALYSIS

We now conduct an analysis of the learned coefficients $\mathbf{c}$ for the pendulum problem in Example 1. For convenience, we present the learned coefficients again, in Table 7. Our analysis consists of two parts: (i) stability, and, (ii) consistency.

|  | $c_1$ | $c_2$ | $c_3$ | $c_4$ | $c_5$ |
|---|---|---|---|---|---|
| $o = 2$ | 2 | -1 | – | – | – |
| $o = 3$ | 1.4 | 0.2 | -0.6 | – | – |
| $o = 4$ | 0.975 | 0.675 | -0.25 | -0.4 | – |
| $o = 5$ | -0.08 | 1.68 | 0.153 | 0.006 | -0.759 |

Table 7: The learned coefficients $\mathbf{c}$ with a varying order $o \in \{2, 3, 4, 5\}$ when solving Example 1.

**Stability analysis.** Following the derivations in Theorem 1 proof presented in Appendix E, we examine the root conditions of the learned coefficients. In Table 8 we report the absolute values of the characteristic polynomial with the coefficients from Table 7. We note that for orders $2, 3$, and $4$, stability is obtained. However, for $o = 5$ the coefficients have one unstable mode. This result suggests that one should not use $o = 5$ for the pendulum problem in Example 1. However, orders lower than 5 yield stability. We believe that a stable fifth-order model is possible to be learned from the data, however, the incorporation of the root condition to the learning process requires adding constraints to the learning process and therefore is beyond the scope of this work.

|  | $|r_1|$ | $|r_2|$ | $|r_3|$ | $|r_4|$ | $|r_5|$ |
|---|---|---|---|---|---|
| $o = 2$ | 1 | 1 | – | – | – |
| $o = 3$ | 1 | 0.6 | 1 | – | – |
| $o = 4$ | 1 | 1 | 0.629 | 0.629 | – |
| $o = 5$ | 1 | 0.73 | 0.73 | 1.4 | 1 |

Table 8: The absolute value of the roots $r_1, \ldots, r_5$ of the characteristic polynomial with coefficients from Table 7.

**Consistency analysis.** There are two conditions that verify the consistency of the learned coefficients. The first condition requires that the sum of the coefficients $\mathbf{c}$ equals to 1, which implies that if the spatial term in Equation (15), the future state (node features) is equal to a weighted average of the previous $o - 1$ states. This implies that a constant solution can always be achieved. The second condition requires that the application of the coefficients $\mathbf{c}$ to a known discrete function yields a consistent approximation to its derivatives of some order.

Note that condition (i) is satisfied by our construction of $\mathbf{c}$. The second condition can be verified numerically, as we show now. To this end, we discretize the function $y = \sin(2\pi t)$ in the interval $[0, 1]$. We then apply the stencils based on the learned coefficients $\mathbf{c}$ as shown in Table 7. As can

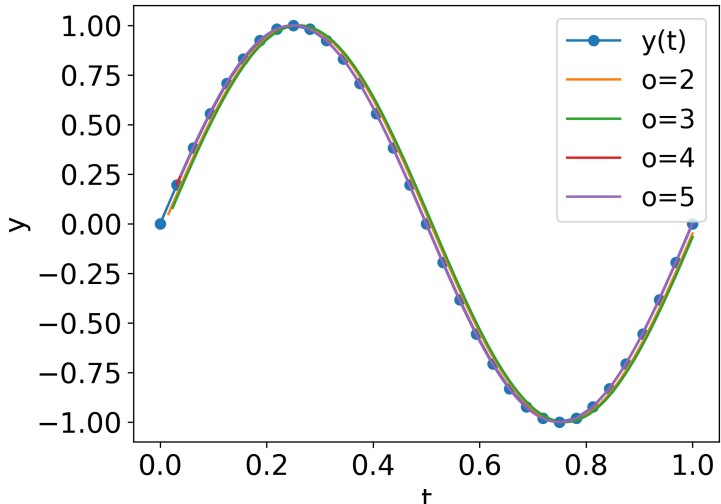

Figure 5: Examining the consistency of the learned coefficients. The learned coefficients model a second-derivative of the test function $y(t) = \sin(2\pi t)$.

be depicted in Figure 5, the application of the stencils to the discrete function $y(t)$ yields a scaled version of the second derivative of $y(t)$ (which also equals to $y$, that is, $\frac{\partial^2 y(t)}{\partial t^2} = \beta y(t)$). This result shows that our TDE-GNN is able to reveal the true order that describes the pendulum's motion, which is a second-order process.

