# OpenReview forum: "Data-Driven Higher Order Differential Equations Inspired Graph Neural Networks"
_ICLR.cc/2024/Workshop/AI4DiffEqtnsInSci — AI4DiffEqtnsInSci @ ICLR 2024 Oral_

### Official Review · Reviewer_Sy8E · 2024-02-21
**Very nice, seminal work.**

**Rating:** 10
**Confidence:** 4

**Review:**

Just be mindful about some typos.
For instance, a the end of the first paragraph in the introduction, "to model the of".

---

### Official Review · Reviewer_QVzE · 2024-02-24
**This paper proposes TDE-GNNs for learning high order temporal dynamics using GNNs. The results are promising, which supports the motivation of TDE-GNN and demonstrates its effectiveness.**

**Rating:** 8
**Confidence:** 4

**Review:**

This paper proposes TDE-GNNs for learning high order temporal dynamics using GNNs. TDE-GNNs model the temporal dynamics via neural ODEs, with maximal order >= 1 for modeling longer and more complex dependencies.

Pros
1. This paper is well written and easy to follow.
2. Extensive empirical studies on various datasets and comparisons to baselines demonstrate the effectivenss of proposed TDE-GNN.
3. The ablation on order $o$ shows that the performance consistently improve as $o$ increase, which well supports the motivation of TDE-GNN.
4. Implementation details are provided.

Cons
1. It seems that with such discretization in Eq(6), it is hard for TDE-GNN to incorporate advantages from neural ODEs, i.e., getting a continuous solution trajectory can be difficult.

---

### Meta-Review · Area_Chair_7KwV · 2024-03-01

**Recommendation:** Accept (Oral)

**Metareview:**

Excellent work that proposes TDE-GNNs for learning high order temporal dynamics using GNNs. TDE-GNNs model the temporal dynamics via neural ODEs, with maximal order >= 1 for modeling longer and more complex dependencies. The paper contains extensive details on the methods and is very well written.

---

### Decision · Program_Chairs · 2024-03-02

Accept (Oral)